# ReLIC: A Recipe for 64k Steps of In-Context Reinforcement Learning for Embodied AI

## Abstract

Intelligent embodied agents need to quickly adapt to new scenarios by integrating long histories of experience into decision-making. For instance, a robot in an unfamiliar house initially wouldn't know the locations of objects needed for tasks and might perform inefficiently. However, as it gathers more experience, it should learn the layout of its environment and remember where objects are, allowing it to complete new tasks more efficiently. To enable such rapid adaptation to new tasks, we present ReLIC, a new approach for in-context reinforcement learning (RL) for embodied agents. With ReLIC, agents are capable of adapting to new environments using 64,000 steps of in-context experience with full attention while being trained through self-generated experience via RL. We achieve this by proposing a novel policy update scheme for on-policy RL called "partial updates" as well as a Sink-KV mechanism that enables effective utilization of a long observation history for embodied agents. Our method outperforms a variety of meta-RL baselines in adapting to unseen houses in an embodied multi-object navigation task. In addition, we find that ReLIC is capable of few-shot imitation learning despite never being trained with expert demonstrations. We also provide a comprehensive analysis of ReLIC, highlighting that the combination of large-scale RL training, the proposed partial updates scheme, and the Sink-KV are essential for effective in-context learning.

## 1 Introduction

A desired capability of intelligent embodied agents is to rapidly adapt to new scenarios through experience. An essential requirement for this capability is integrating a long history of experience into decision-making to enable an agent to accumulate knowledge about the new scenario that it is encountering. For example, a robot placed in an unseen house initially has no knowledge of the home layout and where to find objects. The robot should leverage its history of experiences of completing tasks in this new home to learn the home layout details, where to find objects, and how to act to complete tasks successfully.

To achieve adaptation of decision-making to new tasks, prior work has leveraged a technique called in-context reinforcement learning (RL) where an agent is trained with RL to utilize past experience in an environment (Wang et al., 2016; Team et al., 2023; Duan et al., 2016; Grigsby et al., 2023; Melo, 2022). By using sequence models over a history of interactions in an environment, these methods adapt to new scenarios by conditioning policy actions on this context of interaction history without updating the policy parameters. While in-context RL has demonstrated the ability to scale to a context length of a few thousand agent steps (Team et al., 2023; Grigsby et al., 2023), this falls short of the needs of embodied AI where single tasks by themselves can span thousands of steps (Szot et al., 2021). As a result, the agent cannot learn from multiple task examples because the context required for multiple tasks cannot be accommodated within the policy context. Furthermore, prior work typically focuses on non-visual tasks (Grigsby et al., 2023; Melo, 2022; Ni et al., 2023), where larger histories are easier to incorporate due to the compact state representation.

In this work, we propose a new algorithm for in-context RL, which enables effectively utilizing and scaling to 64,000 steps of in-context experience in partially observable, visual navigation tasks. Our proposed method called Reinforcement Learning In Context (ReLIC), achieves this by leveraging a novel update and data collection technique for training with long training contexts in on-policy RL. Using a long context for existing RL algorithms is prohibitively sample inefficient, as the agent

Figure 1: Overview of the ReLIC approach and problem setup. ReLIC learns a "pixels-to-actions" policy from reward alone via reinforcement learning capable of in-context adapting to new tasks at test time. The figure shows the trained ReLIC policy finding objects in an unseen house. In earlier episodes, the agent randomly explores to find the small target object since the scene is new. But after 64k steps of visual observations, ReLIC efficiently navigates to new target objects.

must collect an entire long context of experience before updating the policy. In addition, the agent struggles to utilize the experience from long context windows due to the challenge of learning long-horizon credit assignment and high-dimensional visual observations. To address this problem, we introduce "partial updates" where the policy is updated multiple times within a long context rollout over increasing context window lengths. We also introduce Sink-KV to further increase context utilization by enabling more flexible attention over long sequences by adding learnable *sink key and value* vectors to each attention layer. These learned vectors are prepended to the input's keys and values in the attention operation. Sink-KV stabilizes training by enabling the agent to not attend to low information observation sequences.

We test ReLIC in a challenging indoor navigation task where an agent in an unseen house operating only from egocentric RGB perception must navigate to up to 80 *small* objects in a row, which spans tens of thousands steps of interactions. ReLIC is able to rapidly in-context learn to improve with subsequent experience, whereas state-of-the-art in-context RL baselines struggle to perform any in-context adaptation. We empirically demonstrate that partial updates and Sink-KV are necessary components of ReLIC. We also show it is possible to train ReLIC with 64k context length. Surprisingly, we show ReLIC exhibits emergent few-shot imitation learning and can learn to complete new tasks from several expert demonstrations, despite only being trained with RL and never seeing expert demonstrations (which vary in distribution from self-generated experiences) during training. We find that ReLIC can use only a few demonstrations to outperform self-directed exploration alone. In summary, our contributions are:

1. We propose ReLIC for scaling in-context learning for online RL, which adds two novel components of partial updates and Sink-KV. We empirically demonstrate that this enables in-context adaptation of over 64k steps of experience in visual, partially observable embodied AI problems, whereas baselines do not improve with more experience.
2. We demonstrate ReLIC is capable of few-shot imitation learning despite only being trained with self-generated experience from RL.
3. We empirically analyze which aspects of ReLIC are important for in-context learning and find that sufficient RL training scale, partial updates, and the Sink-KV modification are all critical.

## 2 RELATED WORK

**Meta RL.** Prior work has explored how agents can learn to quickly adapt to new scenarios through experience. Meta-RL deals with how agents can learn via RL to quickly adapt to new scenarios such as new environment dynamics, layouts, or task specifications. Since Meta-RL is a large space, we only

focus on the most relevant Meta-RL variants and refer the readers to Beck et al. (2023) for a complete survey of Meta-RL. Some Meta-RL works explicitly condition the policy on a representation of the task and adapt by inferring this representation in the new setting (Zhao et al., 2020; Yu et al., 2020; Rakelly et al., 2019). Our work falls under the "in-context RL" Meta-RL paradigm where the policies implicitly infer the context by taking an entire history of interactions as input. $RL^2$ Duan et al. (2016) trains an RNN that operates over a sequence of episodes with RL and the agent implicitly learns to adapt based on the RNN hidden state. Other works leverage transformers for this in-context adaptation (Team et al., 2023; Melo, 2022; Laskin et al., 2022). Raparthy et al. (2023); Lee et al. (2023) also address in-context learning for decision making, but do so via supervised learning from expert demonstrations, whereas our work only requires reward. Most similar to our work is AMAGO (Grigsby et al., 2023), an algorithm for in-context learning through off-policy RL. AMAGO modifies a standard transformer with off-policy loss to make it better suited for long-context learning, with changes consisting of: a shared actor and critic network, using Leaky ReLU activations, and learning over multiple discount factors. Our work does not require these modifications, instead leveraging standard transformer architectures, and proposes a novel update scheme and Sink-KV for scaling the context length with on-policy RL. Empirically, we demonstrate our method scaling to $8\times$ longer context length and on visual tasks, whereas AMAGO focuses primarily on state-based tasks.

**Scaling context length.** Another related area of research scaling the context length of transformers. Prior work extend the context length using a compressed representation of the old context, either as a recurrent memory or a specialized token (Dai et al., 2019; Munkhdalai et al., 2024; Zhang et al., 2024). Other work address the memory and computational inefficiencies of the attention method by approximating it (Beltagy et al., 2020; Wang et al., 2020a) or by doing system-level optimization (Dao, 2023). Another direction is context extrapolation at inference time either by changing the position encoding (Su et al., 2023; Press et al., 2022) or by introducing attention sink (Xiao et al., 2023). Our work utilizes the system-level optimized attention (Dao, 2023) and extends attention sinks for on-policy RL in Embodied AI.

**Embodied AI.** Prior work in Embodied AI has primarily concentrated on the single episode evaluation setting, where an agent is randomly initialized in the environment at the beginning of each episode and is tasked with taking the shortest exploratory path to a single goal specified in every episode (Wijmans et al., 2019; Yadav et al., 2022). In contrast, Wani et al. (2020) introduced the multi-ON benchmark, which extends the complexity of the original task by requiring the agent to navigate to a series of goal objects in a specified order within a single episode. Here, the agent must utilize information acquired during its journey to previous goals to navigate more efficiently to subsequent locations. Go to anything (GOAT) (Chang et al., 2023), extended this to the multi-modal goal setting, providing a mix of image, language, or category goals as input. In comparison, we consider a multi-episodic setting where the agent is randomly instantiated in the environment after a successful or failed trial but has access to the prior episode history.

## 3 METHOD

We introduce Reinforcement Learning In Context (ReLIC) which enables agents to in-context adapt to new episodes without any re-training. ReLIC is built using a transformer policy architecture that operates over a long sequence of multi-episode observations and is trained with online RL. The novelty of ReLIC is changing the base RL algorithm to more frequently update the policy with increasingly longer contexts within a policy rollout and adding Sink-KV to give the model the ability to avoid attending to low-information context. Section 3.1 provides the general problem setting of adapting to new episodes. Section 3.2 details the transformer policy architecture. Section 3.3 describes the novel update scheme of ReLIC. Finally, Section 3.4 goes over implementation details.

### 3.1 PROBLEM SETTING

We study the problem of adaptation to new scenarios in the formalism of meta-RL (Beck et al., 2023). We have a distribution of training POMDPs $\mathcal{M}_i \sim p(\mathcal{M})$, where each $\mathcal{M}_i$ is defined by tuple $(\mathcal{S}_i, \mathcal{S}_i^0, \mathcal{O}_i, \mathcal{A}, \mathcal{T}, \gamma, \mathcal{R}_i)$ for observations $\mathcal{O}_i$, states $\mathcal{S}_i$ which are not revealed to the agent, starting state distribution $\mathcal{S}_i^0$, action space $\mathcal{A}$, transition function $\mathcal{T}$, discount factor $\gamma$, and reward $\mathcal{R}_i$. In our setting, the states, observations, and reward vary per POMDP, while the action space, discount factor, and transition function is shared between all POMDPs.

From a starting state $s_0 \sim \mathcal{S}_i^0$, a policy $\pi$, mapping observations to a distribution over actions, is rolled out for an *episode* which is a sequence of interactions until a maximum number of timesteps, or a stopping criteria. We refer to a *trial* as a sequence of episodes within a particular $\mathcal{M}_i$. The objective is to learn a policy $\pi$ that maximizes the expected return of an episode. At test-time the agent is evaluated on a set of holdout POMDPs.

## 3.2 ReLIC Policy Architecture

Similar to prior work (Grigsby et al., 2023; Team et al., 2023), ReLIC implements in-context RL via a transformer sequence model that operates over a history of interactions spanning multiple episodes. At step $t$ within a trial, ReLIC predicts current action $a_t$ based on the entire sequence of previous observations $o_1, \ldots, o_t$ which may span multiple episodes. In the embodied AI settings we study, the observation $o_t$ consists of an egocentric RGB observation from the robot's head camera along with proprioceptive information and a specification of the current goal. Each of these observation components are encoded using a separate observation encoding network, and the embeddings are concatenated to form a single observation embedding $e_t$. A causal transformer network (Vaswani et al., 2023) $h_\theta$ inputs the sequence of embeddings $h_\theta(e_1, \ldots, e_t)$. From the transformer output, a linear layer then predicts the actions.

The transformer model $h_\theta$ thus bears the responsibility of in-context learning by leveraging associations between observations within a trial. This burden especially poses a challenge in our setting of embodied AI since the transformer must attend over a history of thousands of egocentric visual observations. Subsequent visual observations are highly correlated, as the agent only takes one action between observations. Knowing which observations are relevant to attend to in deciding the current action is thus a challenging problem. In this work, we build our architecture around full attention transformers using the same architecture as the LLaMA language model (Touvron et al., 2023), but modify the number of layers and hidden dimension size to appropriately reduce the parameter count for our setting.

We also introduce an architectural modification to the transformer called **Sink-KV** to improve the transformer's ability to attend over a long history of visual experience from an embodied agent. Building off the intuition that learning to attend over a long sequence of visual observations is challenging, we introduce additional flexibility into the core attention operation by prepending the key and value vectors with a per-layer learnable sequence of *sink KV vectors*. Specifically, recall that for an input sequence $X \in \mathbb{R}^{n \times d}$ of $n$ inputs of embedding dimension $d$, the attention operator projects $X$ to keys, queries and values notated as $K, Q, V$ respectively and all elements of $\mathbb{R}^{n \times d}$ where we assume all hidden dimensions are $d$ for simplicity. The standard attention operation computes softmax $\left( \frac{QK^\top}{\sqrt{d}} \right) V$. We modify calculating the attention scores by introducing learnable vectors $K_s, V_s \in \mathbb{R}^{s \times d}$ where $s$ is the specified number of "sinks". We then prepend $K_s, V_s$ to the $K, V$ of the input sequence before calculating the attention. Note that the output of the attention operation is still $n \times d$, as in the regular attention operation, as the query vector has no added component. We repeat this process for each attention layer of the transformer, introducing a new $K_s, V_s$ in each attention operation. Sink-KV only results in $n_{layers} \times s \times d$ more parameters, which is $0.046\%$ of the 4.5M parameter policy used in this work.

Sink-KV gives the sequence model more flexibility on how to attend over the input. Prior works observe that due to the softmax in the attention, the model is forced to attend to at least one token from the input (Miller, 2023; Xiao et al., 2023). Sink-KV removes this requirement by adding learnable vectors to the key and value. In sequences of embodied visual experiences, this is important as attention heads can avoid attending over any inputs when there is no new visual information in the current observations. This flexibility helps the agent operate over longer sequence lengths.

The calculation of the attentions scores $S$ using the Softmax forces the tokens to attend to values $V$, even if all available values do not hold any useful information, since the sum of the scores is 1 (Miller, 2023). This is especially harmful in cases where the task requires exploration. As the agent explores more, a more useful information may appear in the sequence. If the agent is forced to attend to low information tokens at the beginning of the exploration, it will introduce noise to the attention layers.

---

**Algorithm 1:** Partial Update Pseudocode

1 Define number of steps in trial $T$, number of partial updates $K$, step rollout storage $X_{\text{rollout}}$;
2 **while** *true* **do**
3     Clear the rollout storage;
4     Reset the environment workers;
5     Set $i \leftarrow 0$;
6     **while** $i < K$ **do**
7         Collect $T/K$ environment steps per environment worker and add to $X_{\text{rollout}}$;
8         **if** *rollout storage is full* **then**
9             PPOUpdate($X_{\text{rollout}}$)
10         **end**
11         **else**
12             PPOUpdate($X_{\text{rollout}}[: i \cdot T/K]$);
13             Update KV cache;
14             Shuffle old episodes;
15         **end**
16         $i \leftarrow i + 1$;
17     **end**
18 **end**

---

### 3.3 ReLIC Learning

ReLIC is updated through online RL, namely PPO (Schulman et al., 2017). However, for the agent to be able to leverage a long context window for in-context RL, it must also be trained with this long context window. PPO collects a batch of data for learning by "rolling out" the current policy for a sequence of $T$ interactions in an environment. To operate on a long context window spanning an entire trial, the agent must collect a rollout of data that consists of this entire trial. This is challenging because, in the embodied tasks we consider, we seek to train agents on trials lasting over 64k steps, which consists of at least 130 episodes. As typical with PPO, to speed up data collection and increase the update batch size we use multiple environment workers each running a simulation instance that the policy interacts with in parallel. With 32 environment workers, this corresponds to $\approx 130k$ environment steps between every policy update. PPO policies trained in common embodied AI tasks, such as ObjectNav, have only $128$ steps between updates and require $\approx 50k$ updates to converge (for 32 environment workers, 128 steps per worker between updates and 200M environment steps required for convergence) (Yadav et al., 2022). Executing a similar number of updates would require ReLIC to collect $\approx 6$ billion environment interactions.

ReLIC addresses this problem of sample inefficiency by introducing a *partial update scheme* where the policy is updated multiple times throughout a rollout. First, at the start of a rollout of length $T$, all environment workers are reset to the start of a new episode. Define the number of partial updates as $K$. At step $i \in [0, T]$ in the rollout, the policy is operating with a context length of $i - 1$ previous observations to determine the action at step $i$. Every $T/K$ samples in the rollout, we update the policy. Therefore, at update $N$ within the rollout, the agent has collected $NT/K$ of the $T$ samples in the rollout. The agent is updated using a context window of size $NT/K$, however, the PPO loss is only applied to the final $T/K$ outputs. The policy is changing every $T/K$ samples in the rollout, so the policy forward pass must be recalculated for the entire $NT/K$ window rather than caching the previous $(N-1)T/K$ activations. In the last update in the rollout, after collecting the last $T/K$ steps, we update the policy with the loss applied on all steps in the rollout. We refer to this step as *full update*. At the start of a new rollout, the context window is cleared and the environment workers again reset to new episodes.

### 3.4 Implementation Details

The transformer is modeled after the LLaMA transformer architecture (Touvron et al., 2023) initialized from scratch. Our policy uses a pretrained visual encoder which is frozen during training. A MLP projects the output of the visual encoder into the transformer. We only update the parameters of the transformer and projection layers while freezing the visual encoder since prior work shows this is an

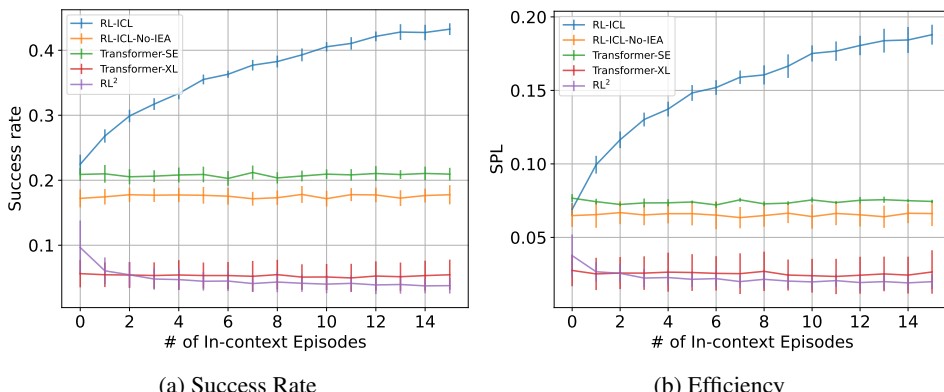

(a) Success Rate    (b) Efficiency

Figure 2: Comparing the in-context learning capability of ReLIC and baselines on EXTOBJNAV. The number of episodes in the trial is displayed on the x-axis. The y-axis displays the success or efficiency at that episode count. Agents capable of in-context learning will increase in success and efficiency when encountering more episodes. Each method is run for 3 random seeds and evaluated on 10k distinct sequences. Error bars are standard deviations over trial outcomes between the 3 seeds.

effective strategy for embodied AI (Khandelwal et al., 2022; Majumdar et al., 2023). For faster policy data collection, we store the transformer KV cache between rollout steps. To fit long context during the training in limited size memory, we used low-precision rollout storage, gradient accumulation (Huang et al., 2019) and flash-attention (Dao, 2023). After each policy update, we shuffle the older episodes in the each sequence and update the KV-cache. Shuffling the episode serves as regularization technique since the agent sees the same task for a long time. It also reflects the lack of assumptions about the order of episodes, an episode should provide the same information regardless of whether the agent experiences it at the beginning or at the end of the trial.

We use the VC-1 visual encoder and with the ViT-B size (Majumdar et al., 2023). We found the starting VC-1 weights performed poorly at detecting small objects, which is needed for the embodied AI tasks we consider. We therefore finetuned VC-1 on a small objects classification task. All baselines use this finetuned version of VC-1. We provide more details about this VC-1 finetuning in Appendix D and details about all hyperparameters in Appendix B.2.

## 4 EXPERIMENTS

We first introduce the Extended Object Navigation (EXTOBJNAV) task we use to study in-context learning for embodied navigation. Next, we analyze how ReLIC enables in-context learning on this task and outperforms prior work and baselines. We then analyze ablations of ReLIC and analyze its behaviors. We also show ReLIC is capable of few-shot imitation learning. Finally, we show that ReLIC outperforms the methods in Lee et al. (2023) on the existing Darkroom and Miniworld tasks.

### 4.1 EXTOBJNAV: EXTENDED OBJECT NAVIGATION

To evaluate ICL capabilities for embodied agents, we introduce EXTOBJNAV, an extension of the existing Object Navigation (OBJECTNAV) benchmark. EXTOBJNAV assesses an agent's ability to find a sequence of objects in a house while operating from egocentric visual perception. For each object, the agent is randomly placed in a house and must locate and navigate to a specified object category. The agent used is a Fetch robot equipped with a $256 \times 256$ RGB head camera. Additionally, the agent possesses an odometry sensor to measure its relative displacement from the start of the episode. Navigation within the environment is executed through discrete actions: move forward 0.25 meters, turn left or right by 30 degrees, and tilt the camera up and down by 30 degrees. The agent also has a stop action, which ends the episode.

EXTOBJNAV uses scenes from the Habitat Synthetic Scenes Dataset (HSSD) (Khanna et al., 2023) along with a subset of the YCB object dataset (Calli et al., 2015) containing 20 objects types. Note that EXTOBJNAV requires navigating to *small* objects unlike other OBJECTNAV variants that use

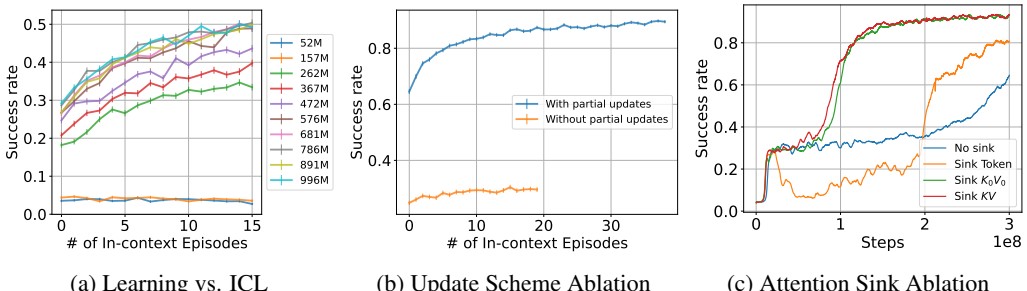

(a) Learning vs. ICL       (b) Update Scheme Ablation       (c) Attention Sink Ablation

Figure 3: Analyzing ReLIC ICL capabilities. Fig. 3a shows increased RL training results in agents that have a higher base success and stronger ICL capabilities with error bars giving standard error on the evaluation episodes. Fig. 3b shows the partial updates are important in ReLIC. Fig. 3c shows Sink-KV is important for learning speed and stability. The results in Fig.3b,3c use the smaller ReplicaCAD scenes for easier analysis and thus have higher overall success rates. These results performed on the easier ReplicaCAD scenes to save compute, so the numbers are higher overall.

large receptacles as goals (Team, 2021). This allows us to increase the dataset diversity by sampling objects randomly in the environment, unlike OBJECTNAV, where the receptacles are fixed parts of the scanned meshes. The random sampling also precludes the agent from using priors over object placements in scenes, forcing it to rely on the experience in its context.

EXTOBJNAV defines a *trial* as a sequence of episodes within a fixed home layout, where a home layout is defined by a combination of a floorplan, a furniture layout and set of object placements. A home layout contains on an average 22 objects where multiple object instances may be of the target category. Within an episode in the trial, a target object category is randomly selected and the agent is randomly placed in the house. The episode is successful if the agent calls the stop action within 2 meters of the object, with at least 10 pixels of the object in the current view. If the object is not found within $500$ steps the episode counts as a failure.

We evaluate the agents on unseen scenes from HSSD and report the success rate (SR) and Success-weighted by Path Length (SPL) metrics (Anderson et al., 2018). Specifically, we look at the SR and SPL of an agent as it accumulates more episodes in-context. Ideally, with more in-context episodes within a home layout, it should be more adept at finding objects and its SR and SPL should improve. See Appendix A for further details on the EXTOBJNAV.

## 4.2 IN-CONTEXT LEARNING ON EXTOBJNAV

In this section, we compare the ability of ReLIC and baselines to in-context learn in a new home layout. We compare ReLIC to the following baselines:

- **RL**[2] Duan et al. (2016): Use an LSTM and keep the hidden state between trial episodes.
- **Transformer-XL (TrXL)** Dai et al. (2019): Use Transformer-XL and updates the constant-size memory recurrently. This is the model used in Team et al. (2023) trained in our setting. Following Team et al. (2023) we use PreNorm (Parisotto et al., 2019) and gating in the feedforward layers (Shazeer, 2020).
- **ReLIC-No-IEA**: ReLIC without Inter-Episode Attention (IEA). Everything else, including the update scheme is the same as ReLIC.
- **Transformer-SE**: A transformer-based policy operating over only a single episode (SE) and without the update schemes from ReLIC.

All baselines are trained for 500M steps using a distributed version of PPO (Wijmans et al., 2019). Methods that utilize multi-episode context are trained with a context length of 4k, and use 8k context length during inference (unless mentioned otherwise, e.g. in our long-context experiments). The results in Figure 2 demonstrate ReLIC achieves better performance than baselines on 8k steps of ICL, achieving 43% success rate v.s. 22% success rate achieved by the closest performing baseline (Transformer-SE).

Additionally, ReLIC is able to effectively adapt to new home layouts throughout the course of the trial. In the first episode of the trial, transformer-based baseline methods attain a similar base performance

of around $20\%$ success rate. However, as more episodes arrive, the performance of ReLIC increases. The recurrent models, Transformer-XL and $RL^2$, have lower base performance at $10\%$ success rate and show no in-context learning. The performance of $RL^2$ degrades with more in-context episodes, which is aligned with the inability of the LSTM to model long sequences.

After 15 episodes of in-context experience, the success rate of ReLIC increases from $23\%$ to $43\%$. The baselines do not possess this same ICL ability and maintain constant performance with subsequent in-context episodes. ReLIC also in-context learns to navigate faster to objects, as measured by the gap in SPL. As the trial progresses, the agent is able to more efficiently navigate to objects in the house with the SPL of ReLIC increasing from $0.07$ to $0.188$. The baselines are unable to improve efficiency in-context and maintain a SPL of $0.025$ to $0.075$ throughout the entire trial.

### 4.3 ReLIC Ablations and Analysis

We demonstrate that the partial updates in ReLIC and Sink-KV are crucial to learning with RL over long context windows and acquiring ICL capabilities. We run these ablations in the smaller ReplicaCAD (Szot et al., 2021) scenes to make methods faster to train, but other details of the task remain the same. We then show that ICL emerges later in the training and the context length in ReLIC can be even further increased.

**No Partial Updates.** Firstly, we remove the partial updates in ReLIC and find that it performs poorly (Figure 3b), achieving $40\%$ lesser SR at the first episode. This model also shows little ICL abilities with the SR only increasing $5\%$ by the end of the trial versus a $25\%$ increase when using partial updates.

**Sink-KV.** Next, we demonstrate that using Sink-KV is necessary for sample-efficient in-context RL learning. We trained the model on ReplicaCAD with and without attention sinks. The learning curves in Figure 3c shows that learning is more stable and faster with Sink-KV which achieves 90% success rate at 200M steps. It also shows that Sink-KV performs similar to Softmax One (Miller, 2023), referred to as Sink $K_0V_0$. Without attention sink mechanisms, learning is slow and achieves less than 40% success rate after 200M steps and reaching 64% at 300M steps. Using sink token (Xiao et al., 2023), the training becomes unstable, achieving 40% success rate at 200M steps training and reaching 80% success rate at 300M steps. The details of the different sink attentions, their implementations and how the attention heads use the Sink-KV can be found in Appendix E.

**Training Steps v.s. ICL Abilities.** We find that ReLIC only acquires ICL capabilities after sufficient RL training. As demonstrated in Figure 3a, the agent is only capable of ICL after 157M steps of training. Models trained for 52M and 157M remain at constant success with more in-context experience. Further training does more than just increase the base agent performance in the first episode of the trial. From 262M steps to 367M steps, the agent base performance increases by $2\%$, yet the performance after 15 episodes of ICL performance increases $10\%$. This demonstrates that further training is not only improving the base capabilities of the agent to find objects, but also improving the agent's ability to utilize its context across long trials spanning many episodes.

**Context length generalization.** Next, we push the abilities of ReLIC to in-context learn over contexts much larger than what is seen during training. In this experiment, we evaluate ReLIC model, trained with 4k context length, on 32k steps of experience, which is enough to fit 80 episode trials in context. Assuming that the simulator is operating at 10Hz, this is almost 1 hour of agent experience within the context window. Note that for this experiment, we use our best checkpoint, which is trained for 1B steps. The results demonstrate that ReLIC can generalize to contexts $8\times$ larger at inference. Figure 4a shows ReLIC is able to further increase the success rate to over $55\%$ after 80 in context episodes and consistently maintains performance above $50\%$ after 20 in-context episodes.

**64k steps trials.** Finally, we investigate scaling *training* ReLIC with 64k context length. We use the same hyperparameters as Section 4.2, but increase the number of partial updates per rollout such that the policy is updated every 256 steps, the same number of steps used in ReLIC. Fig. 4b shows that the model can in-context learn over 175 episode and continue to improve success rate. More details are available at Appendix C.5.

In Appendix C.1 we analyze the performance of ReLIC per object type. In Appendix C.2 we qualitatively analyze what the agent attends to in successful and failure episodes. Finally, in Appendix C.3, we show that not shuffling episodes in the context during training leads to worse performance.

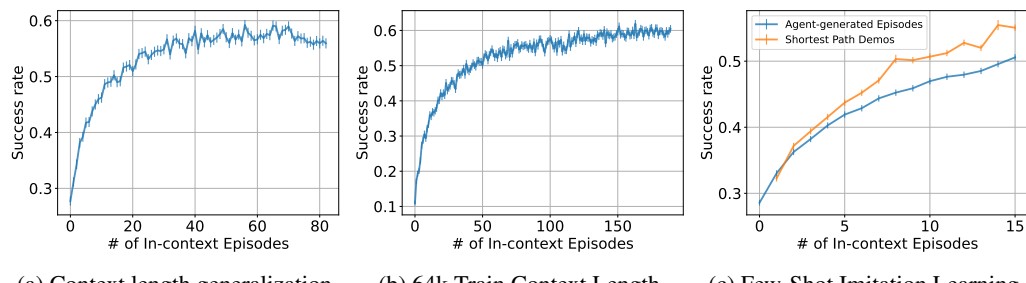

(a) Context length generalization    (b) 64k Train Context Length    (c) Few-Shot Imitation Learning

Figure 4: (a) ReLIC trained with context length 4k generalizes to operating at 32k steps of in context experience in a new home layout. (b) ReLIC trained at 64k context length shows ICL abilities over 175 episodes. (c) ReLIC can do few-shot imitation learning despite not training for it. The error bars represent the standard error.

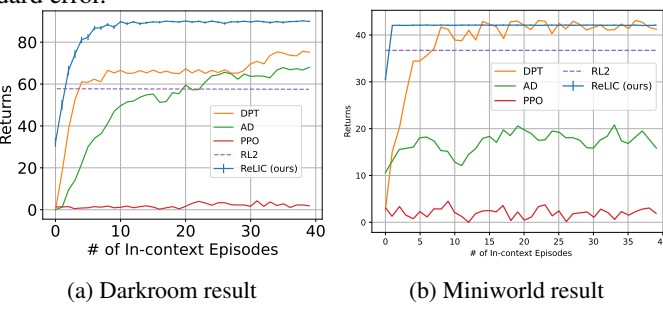

(a) Darkroom result    (b) Miniworld result

Figure 5: ICL comparison of ReLIC and baselines in the Darkroom and Miniworld tasks. ReLIC has a higher base performance and adapts to new tasks with less experience. The baselines numbers are obtained from Figures 4b,d of Lee et al. (2023). Error bars are the standard error of the evaluation results computed over 2k sequences.

## 4.4 Emergent Few-Shot Imitation Learning

In addition to learning in-context from self-generated experience in an environment, ReLIC can also use its context to learn from demonstrations provided by an external agent or expert, despite never being trained on demonstrations and only learning from self-generated experience. We consider the setting of few-shot imitation learning (Duan et al., 2017; Wang et al., 2020b) where an agent is given a set of trajectories $\{\tau_1, \ldots, \tau_N\}$ demonstrating behavior reaching desired goals $\{g_1, \ldots, g_N\}$. The agent must then achieve a new $g_{N+1}$ in new environment configurations using these demonstrations. ReLIC is able to few-shot imitation learn by taking the expert demonstration as input via the context. Specifically, we generate $N$ expert shortest path trajectories navigating to random objects from random start positions in an unseen home layout. The success rate of these demos is around $80\%$ due to object occlusions hindering the shortest path agent from viewing the target object which is required for success. These $N$ trajectories are inserted into the context of ReLIC and the agent is instructed to navigate to a new object in the environment.

In Figure 4c we show that ReLIC can utilize these expert demonstrations despite never seeing such shortest paths during training. Figure 4c shows the success rate of ReLIC in a single episode after conditioning on some number of shortest path demonstrations. More demonstrations cover more of the house and the agent is able to improve navigation success. We also compare to the success rate of an agent that has $N$ episodes of experience in the house as opposed to $N$ demonstrations. Using the demonstrations results in better performance with $5\%$ higher success rate for $N = 16$.

## 4.5 Darkroom and Miniworld

In this section, we evaluate ReLIC on the Darkroom (Zintgraf et al., 2020) and Miniworld (Chevalier-Boisvert, 2018) environments and compare to the results from Lee et al. (2023) to provide a comparison with existing baselines on these simpler benchmarks. We directly take the numbers from Lee et al. (2023) which include Decision-Pretrained Transformer (DPT), a supervised pretraining method for in-context meta-RL, Algorithm Distillation (AD) (Laskin et al., 2022), Proximal Policy Optimization (PPO) and RL². ReLIC is trained with context length 512, which fits 10 Miniworld and 5 Darkroom episodes. Policies are evaluated with 40 in-context episodes. Note that DPT is trained

with actions from an optimal policy in these environments while ReLIC is not. Full details are in Appendix B.3.

**Darkroom.** Figure 5a shows that ReLIC outperforms all previous methods in the Darkroom task. Specifically, ReLIC achieves base performance of 32 while the other methods have base performance lower than 2 returns. ReLIC reaches 89 returns after 10 in-context episodes which is higher than 75 achieved by DPT after 39 in-context episodes.

**Miniworld.** ReLIC has a higher base performance of 30 episode return compared to the best base performance of 10 as shown in Figure 5b. It quickly reaches 42 returns after just 2 in-context episodes while DPT reaches the same result after 14 in-context episodes. ReLIC also shows stable performance as the number of episodes increase compared to DPT which shows oscillation in the performance. In Appendix C.4, we also show the importance of Sink-KV and partial updates in both tasks.

## 5    Conclusion and Limitations

The ability of an agent to rapidly adapt to new environments is crucial for successful Embodied AI tasks. We introduced ReLIC, an in-context RL method that enables the agent to adapt to new environments by in-context learning with up to 64k environment interactions and visual observations. We studied the two main components of ReLIC: *partial updates* and the *Sink-KV* and showed both are necessary for achieving such in-context learning. We showed that ReLIC results in significantly better performance on a challenging long-sequence visual task compared to the baselines.

Limitations of the approach are that we found for ICL to emerge, it requires a diverse training dataset on which the model can not overfit. There is no incentive for the model to learn to use the context if it can overfit the task. We were able to address that in the dataset generation by creating different object arrangements for each scene which made it challenging for the model to memorize the objects arrangements. Another is that our study only focuses on several environments. Future work can explore this same study in more varied environments such as a mobile manipulation task where an agent needs to rearrange objects throughout the scene. Finally, ReLIC requires large amounts of RL training to obtain in-context learning capabilities. The success of ReLIC in ExtObjNav is also relative low for practical applications. One path to improving this performance is to scale training with more RL training and in-context learning. Figure 4b shows the performance is still improving after 64k steps of in-context experience. Figure 16 also shows that ReLIC is still improving after 1 billion RL steps. Another path is to improve ability to generalize to new scenes by increasing the number of training scenes from the 37 in the HSSD dataset used in ExtObjNav through procedurally generated scenes.

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

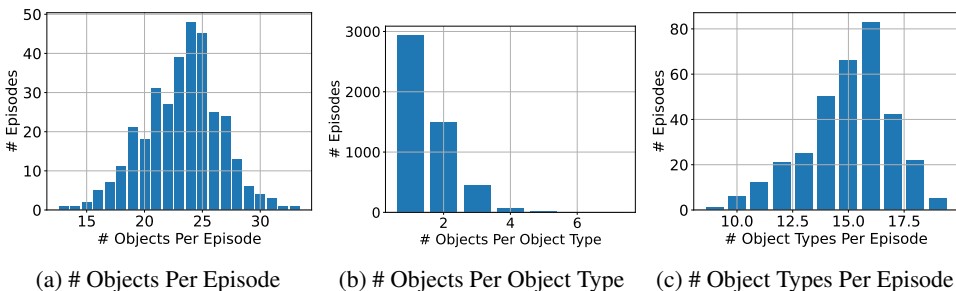

(a) # Objects Per Episode     (b) # Objects Per Object Type     (c) # Object Types Per Episode

Figure 6: The distribution of the objects and object types in the data.

# A   ADDITIONAL EXTOBJNAV DETAILS

The EXTOBJNAV is a small object navigation task. We use the same training (37) and validation (12) scenes as Yenamandra et al. (2023). The data is generated by randomly placing objects from the 20 object types, a subset of the YCB Calli et al. (2015) dataset, on random receptacles. The data is generated by sampling between 30 to 40 object instances and placing them on receptacles, and subsequent filtering of the objects that are not reachable by the agent. The filtering is done by placing the agent in front of the object and evaluating whether the agent can meet the success criteria. If the agent can meet the critera, we retain the object. Otherwise, we discard the object. The distribution of the objects and object types are in Figure 6.

The reward function is defined as follows:

- Change in geodesic distance to the closest object $r_d = -\Delta d$ where $d$ is the geodesic distance to the closest object. The closest object can change across the episode.
- Slack reward of $-0.001$.
- Succes reward of 2.

The episode is considered a success if the agent selects the *Stop* action while it is within 2 meters of an instance of the target type and has 10 pixels of this instance in the view.

# B   ADDITIONAL METHOD DETAILS

## B.1   MODEL TRAINING

In this section, we discuss the training setup for ReLIC experiment.

**The workers and the batch size.** We use 20 environment workers per GPU. Since we use 4 GPUs in parallel, there are 80 environment workers in total. The micro batch size is 1 and we accumulate the gradient for 10 micro batches on the 4 GPUs which makes the effective batch size 40.

**RL algorithm.** We use PPO Schulman et al. (2017) to train the model with $\gamma = 0.99$, $\tau = 0.95$, entropy coefficient of 0.1 and value loss coefficient of 0.5.

**Optimizer.** We use Adam Kingma & Ba (2014) optimizer to learn the parameters.

**Learning rate schedule.** We use learning rate warm up in the first 100,000 environment interactions. The learning rate starts with $LR_0 = 2e - 7$ and reaches $LR = 2e - 4$ at the end of the warm up. Cosine decay Loshchilov & Hutter (2017) is used after the warm-up to decay the learning rate to 0 after 1B environment interactions.

**Precision.** We use FP16 precision for the visual encoder and keep the other components of the model as FP32.

**Rollout Storage.** The rollout storage size is 4096. We store the observations and the visual embeddings in rollout in low-precision storage, specifically in FP16 precision.

**Regularization.** We follow Reed et al. (2022) in using depth dropout Huang et al. (2016) with value 0.1 as regularization technique. We also shuffle the in-context episodes after each partial updates.

**Hardware Resources and Training Time.** The model is trained for 1B steps on 4x Nvida A40 for 12 days.

### B.2 HYPERPARAMETERS

We list the hyperparameters for the different experiments discussed in section Section 4.2.

**ReLIC**: The hyperparameters used in ReLIC can be found in Table 1 and the hyperparameters of the transformer model used in the training can be found in Table 1.

**RL$^2$**: For implementing RL$^2$, we build on the default PPO-GRU baseline parameters in Habitat 3.0 Puig et al. (2023). We set the number of PPO update steps to 256, and the hidden size of the GRU to 512. The scene is changed every 4096 steps during training, and the hidden state is reset to zeros after every scene change.

**ReLIC-No-IEA**: We use the same model and hyperparameters as ReLIC. The only difference is that we set the attention mask to restrict the token to only access other tokens within the same episode.

**Transformer-SE**: We use the same model and hyperparameters as ReLIC. However, we limit the training sequence to a fixed size 385 old observations + 256 new observations. The choice of the old number of observations is made such that we never truncate an episode which is at most 500 steps. The attention mask is set to restrict the tokens to only access other tokens in the same episode.

**Transformer-XL (TrXL)** Dai et al. (2019): Use Transformer-XL and update the constant-size memory recurrently. We follow Team et al. (2023) in that we use PreNorm Parisotto et al. (2019) and use gating in the feedforward layers Shazeer (2020). We experiment with two values for the memory size, 256 and 1024, using TrXL without gating and found that the model is able to learn with 256 memory but is unstable with 1024 memory. We use 256 memory size which gives the agent context of size $L \times N_m = 4 \times 256 = 1024$ where $L$ is the number of layers. Except for the memory, we use the same number of layers and heads and the same hidden dimensions as ReLIC.

| Hyperparameter | Value |
|---|---|
| # Layers | 4 |
| # Heads | 8 |
| Hidden dimensions | 256 |
| MLP Hidden dimensions | 1024 |
| # Sink-KV | 1 |
| Attention sink | Sink $KV_0$ |
| Episode index encoding | RoPE Su et al. (2023) |
| Within-episode position encoding | Learnable |
| Activation | GeLU Shazeer (2020) |
| Rollout size | 4096 |
| total # updates per rollout | 16 |
| # partial updates | 15 |
| # full updates | 1 |

Table 1: ReLIC and baseline hyperparameters

### B.3 DARKROOM AND MINIWORLD HYPERPARAMETERS

We use smaller transformer for these two tasks described in Table 2. The ReLIC hyperparameters are provided in Table 2. For the visual encoder, we use the CNN model used in Lee et al. (2023) and train it from scratch. The other hyperparameters are the same as described in Appendix B.2.

## C MORE EXPERIMENTS

The result in Figure 7 shows that the model is able to learn and generalize on 64k sequence length.

| Hyperparameter | Value |
|---|---|
| # Layers | 2 |
| # Heads | 8 |
| Hidden dimensions | 64 |
| MLP Hidden dimensions | 256 |
| # Sink-KV | 1 |
| Attention sink | Sink $K_0 V_0$ |
| Episode index encoding | RoPE Su et al. (2023) |
| Within-episode position encoding | Learnable |
| Activation | GeLU Shazeer (2020) |
| Rollout size | 512 |
| # updates per rollout | 4 (Darkroom), 2 (Miniworld) |
| # partial updates | 3 (Darkroom), 1 (Miniworld) |
| # full updates | 1 |

Table 2: Hyperparameters for ReLIC and baselines in Miniworld and Darkroom.

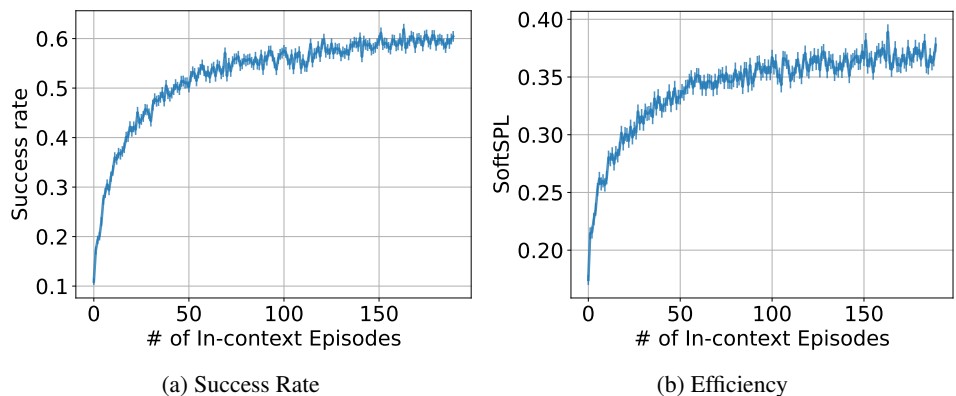

(a) Success Rate         (b) Efficiency

Figure 7: The success and efficiency of training and evaluating ReLIC with 64k context length.

## C.1 ReLIC PER OBJECT TYPE

In Figure 8a, we analyze the ICL performance of ReLIC per object type. Specifically, we specify the same object type target for the agent repeatedly for 19 episodes. Similar to the main experiments, the agent is randomly spawned in the house. As Figure 8a illustrates, ReLIC becomes more capable at navigating to all object types in subsequent episodes. The agent is good at adapting to finding some objects such as bowls, cracker box, and apples. Other objects, such as strawberry and tuna fish can, remain difficult. In Figure 8b, we show that with 19 episodes of ICL, the agent is can reliably navigate to any object type in the house despite having different object types as target in the context. This demonstrates the agent is able to utilize information about other object targets from the context.

## C.2 ANALYZING ATTENTION SCORES

In this section, we show that the agent is able to utilize the in-context information by inspecting the attention scores patterns in the attention heads. We generate the data by letting the agent interact with an unseen environment for 19 episodes which produced a sequence of 2455 steps. A random object type is selected as a target in each episode. By inspecting the attention scores of the attention heads, we found 4 patterns shown in Figure 9.

- **Intra-episode attention**: In this pattern, the agent attends only to the running episode, Figure 9a.
- **Inter-episodes attention**: Inter-episodes attention is where the agent accesses the information from previous episodes, Figure 9b.
- **Episode-invariant attention**: The agent is able to attend to certain tokens which do not change on changing the episode, Figure 9c.

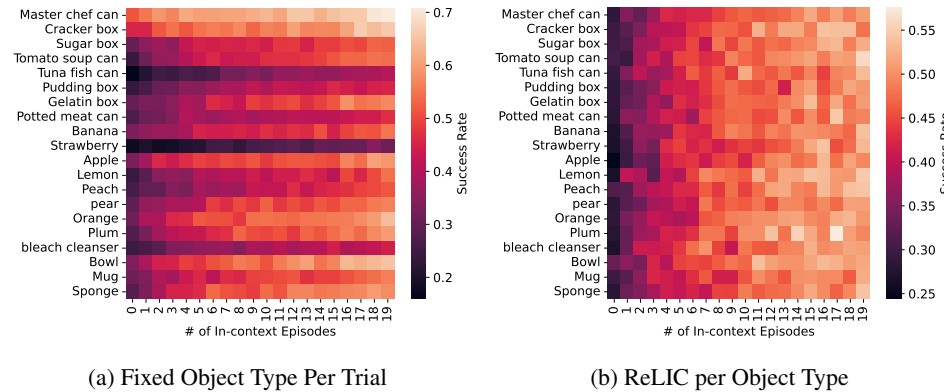

(a) Fixed Object Type Per Trial        (b) ReLIC per Object Type

Figure 8: Analysis of how ReLIC learns to navigate to particular object types through ICL. (a) compares the number of consecutive episodes within a trial an object appears v.s. the success rate. The agent becomes more capable at navigating to that object type for subsequent episodes. (b) shows the episode index within the trial that the object first appears v.s. the average success rate for different objects. As the agent acquires more experience in-context, it can proficiently navigate to any object type.

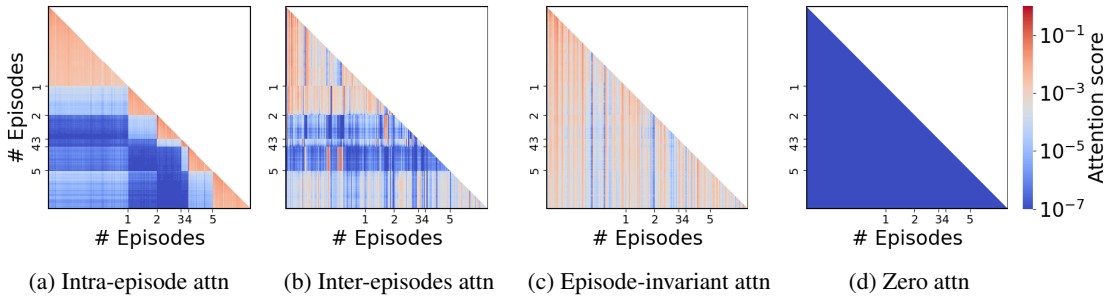

(a) Intra-episode attn    (b) Inter-episodes attn    (c) Episode-invariant attn    (d) Zero attn

Figure 9: Attention scores patterns of a sequence with 1024 steps. We found 4 attention patterns in the heads of a trained policy: (a) Intra-episode attention where the attention head assigns high score to the running episode, (b) Inter-episode attention pattern where the attention head assigns high score to the context, without being constrained to the running episode, (c) the episode-invariant pattern where the attention head attends to the same tokens regardless of the episode structure in the context, and (d) the zero attention pattern where the attention head assign all attention scores to the Sink-KV.

- **Zero attention**: Some heads have 0 attention scores for all tokens which would not be possible with the vanilla attention.

We further analyze the attention pattern between successful and failure episodes. We collect 2455 steps in a trial and then probe the agent's attention scores by querying each object type by adding a new observation with the desired object type at the final step. Figure 14 shows that the agent is able to recall multiple instances of the target object types in its history.

Figure 15 shows the attention scores for all 20 object types when selected in the 1st step of a new episode after 19 episodes.

## C.3   IMPACT OF EPISODE SHUFFLING

We ran ReLIC on ReplicaCAD with and without in-context episodes shuffling. Figure 10 shows that ReLIC marginally suffers at in-context learning (ICL) when not shuffling episodes in the context during training. Specifically, the final ICL performance has a 3% lower success rate and the ICL is less efficient. We believe that shuffling the episodes in the context during the training acts as regularization since it creates diverse contexts.

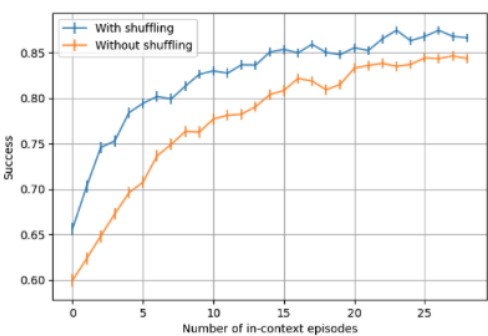

Figure 10: The effect of shuffling in-context episodes during the training

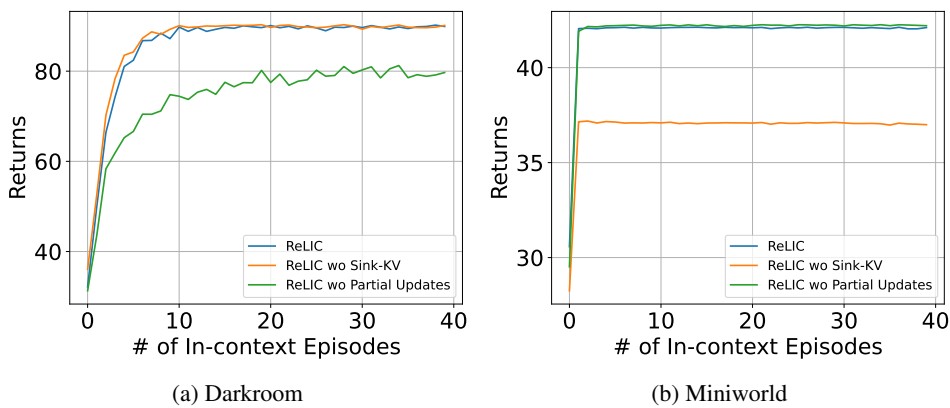

| (a) Darkroom | (b) Miniworld |

Figure 11: Ablating ReLIC components on Darkroom and Miniworld.

## C.4 ABLATIONS IN MINIWORLD AND DARKROOM

In this section, we run the partial udpate and Sink-KV ablations from Section 4.3 on the Darkroom and Miniworld tasks from Section 4.5. The result shows that different components in ReLIC is crucial for different tasks while using ReLIC is as good as or better than ReLIC without its components. The ablation shows that Partial Updates is crucial for long horizon tasks like Darkroom and EXTOBJNAV as shown in Figs. 3b and 11a, which have horizon of 100 and 500 steps respectively, but not important for short horizon tasks like Miniworld, which is 50 steps, as shown in Fig. 11b. It also shows that Sink-KV is important for tasks with rich observations like Miniworld and EXTOBJNAV, which are visual tasks, compared to the Darkroom, which is a grid world task.

## C.5 TRAINING WITH 64K CONTEXT LENGTH

In the main experiment, we showed that we can train on 4k steps and inference for 32k steps. In this experiment, we show that our method ReLIC is able to train with 64k sequence length. We used the same hyperparameters in the main experiment, except the training sequence length which we set to 64k and the number of updates per rollout is increased so that we do updates every 256 steps, same as the main experiment.

## D VISUAL ENCODER FINETUNING

We finetuned the visual encoder on a generated supervised task before freezing it to be used in our experiments. Each sample, Figure 12, in the data is generated by placing the agent in front of a random object then the RGB sensor data is used as input $X$. The output $y$ is a binary vector of size 20, the number of available object types, where each element represents whether the corresponding

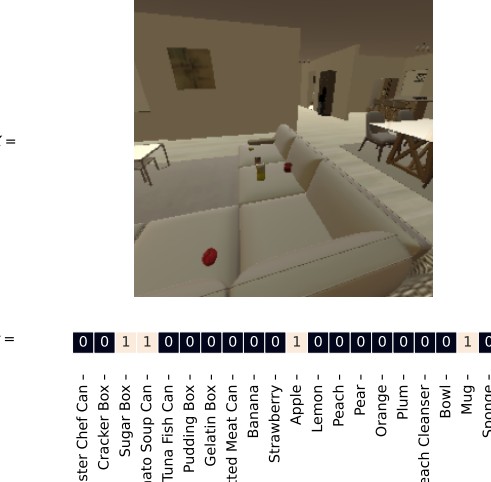

Figure 12: Sample of the finetuning data

object type is in the image or not. The object type is considered in the image if there is an instance of this object in the image with more than 10 pixels. 21k samples are generated from the training scenes and object arrangements. The 21k samples are then split to training and validation data with ratios 90% to 10%.

The VC-1 model is finetuned using the Dice loss Sudre et al. (2017) by adding a classification head to the output of '[CLS]' token using the generated data. The classification head is first finetuned for 5 epochs with $LR = 0.001$ while the remaining of the model is frozen. Then the model is unfrozen and finetuned for 15 epochs with $LR = 0.00002$.

## E    SINK KV

We introduce Sink $KV$, a modification to the attention calculation in the attention layers. We first describe the vanilla attention Vaswani et al. (2023), the issue and the motivation to find a solution. Then we discuss the proposed solutions and introduce the Sink-KV technique. Finally, we anlayze different variants of Sink-KV.

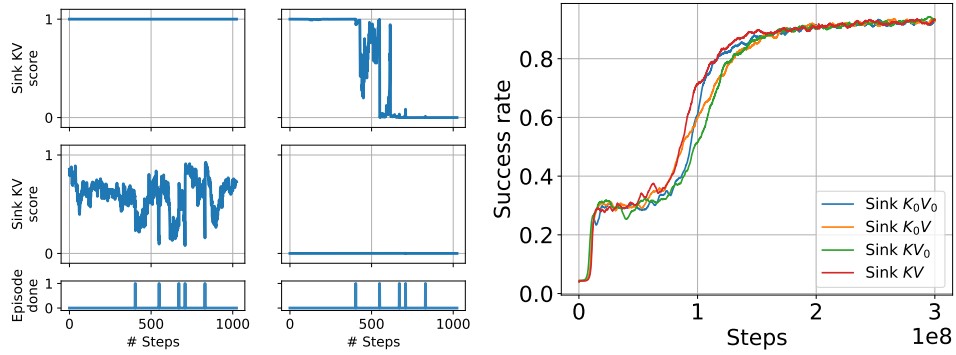

(a) Different patterns of Sink $KV$ scores for 1k input tokens.

(b) The learning curve for the Sink $KV$ variants.

Figure 13: Sink $KV$ analysis.

### E.1 MOTIVATION

The vanilla attention is the component responsible for the interaction between the tokens in the sequence. The output for each token is calculated by weighting the value of all tokens. The input to the attention layer is the embeddings of the input tokens $E \in \mathbb{R}^{n \times d}$ where $n$ is the number of input tokens and $d$ is the dimension of the embeddings.

First the embeddings $E$ are linearly projected to the Key $K$, Value $V$ and Query $Q$. Then the attention scores are calculated using $S = \text{Softmax}(QK^T/\sqrt{d_k})$ where $d_k$ is the dimensions of the keys. The output $A$ is calculated as a weighted sum of the values $V$, $A = SV$.

The calculation of the attentions scores $S$ using the Softmax forces the tokens to attend to values $V$, even if all available values do not hold any useful information, since the sum of the scores is 1 (Miller, 2023). This is especially harmful in cases where the task requires exploration. As the agent explores more, a more useful information may appear in the sequence. If the agent is forced to attend to low information tokens at the beginning of the exploration, it will introduce noise to the attention layers.

### E.2 SOLUTIONS

Softmax One from Miller (2023) addresses this issue by adding 1 to the denominator of the Softmax, $\text{Softmax}_1(x_i) := \exp(x_i)/(1 + \sum_j \exp(x_j))$, which is equivalent to having a token with $k = 0$ and $v = 0$. This gives the model the ability to have 0 attention score to all tokens, we refer to Softmax One as Sink $K_0 V_0$.

Sink tokens from Xiao et al. (2023) are another approach to address the same issue by prepending learnable tokens to the input tokens $E = [E_s \circ E_{input}]$ where $E$ is the input embedding to the model and $[A \circ B]$ indicates concatenation along the sequence dimension of the $A$ and $B$ matrices .

Sink-KV is a generalization of both approaches. It modifies the attention layer by adding a learnable Key $K_s \in \mathbb{R}^{n \times d_k}$ and values $V_s \in \mathbb{R}^{n \times d}$. In each attention layer, we simply prepend the learnable $K_s$ and $V_s$ to the vanilla keys $K_v$ and values $V_v$ to get the $K = [K_s \circ K_v]$ and $V = [V_s \circ V_v]$ used to calculate the attention scores then the attention output.

In the case $K_s = 0$ and $V_s = 0$, Sink-KV becomes equivalent to Softmax One. It can also learn the same $K$s and $V$s corresponding to the Sink Token since our model is casual and the processing of the Sink Token is not affected by the remaining sequence.

### E.3 SINK-KV VARIANTS

We tried a variant of Sink-KV where the either the Value or the Key is set to 0, referred to as Sink $KV_0$ and Sink $K_0 V$ respectively. All variants perform similarly in terms of the success rate as shown in Figure 13b.

Figure 13a shows different patterns the model uses the Sink $KV_0$. The model can assign all attention scores to the Sink $KV_0$, which yields a zero output for the attention head, or assign variable scores at different time in the generation. For example, one the attention heads is turned off during the 1st episode of the trial by assigning all attention score to the Sink $KV_0$ then eventually move the attention to the input tokens in the new episodes. The model is also able to ignore the Sink $KV_0$ by assigning it 0 attention scores as shown in the figure.

## F INFERENCE TIMES

In this section, we compare the the inference speeds of ReLIC, Transformer-XL, and RL2 listed in Table 3. All numbers were obtained with batch size 20 on a single A40 GPU. The models are all about 5.5M parameters in size. Despite all methods operating with the same 8k context length, they all have similar inference speeds with RL2 being faster due to its LSTM rather than transformer based architecture.

|                    | ReLIC  | Transformer-XL | RL2    |
| ------------------ | ------ | -------------- | ------ |
| Actions per Second | 732.14 | 777.25         | 893.73 |

Table 3: Comparison of Actions per Second across ReLIC, Transformer-XL, and RL2.

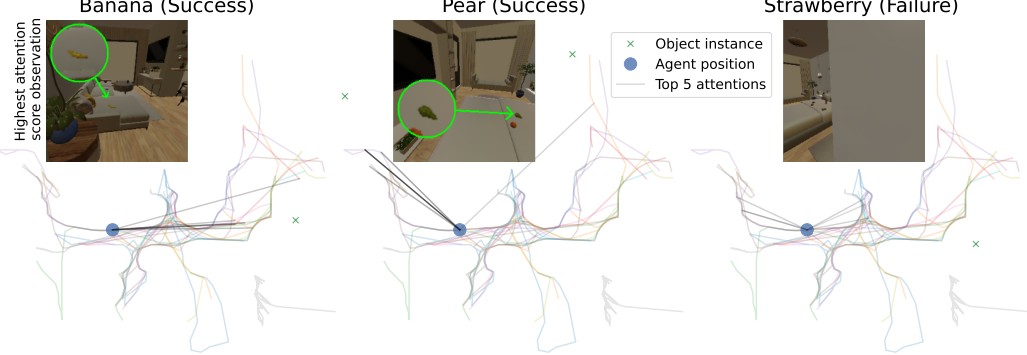

Figure 14: Visualization of an inter-episode attention head, see Appendix C.2. The colored curves are the trajectories of previous episodes. The blue circle is the agent's position. The green Xs are the instances of the target object type. The black lines represent the agent's attention when the target is the object type mentioned above the image. The lines connect the agent with the point in history that it attends to, the opacity of the line represents the attention score. The overlaid image is visual observation with the highest attention score.

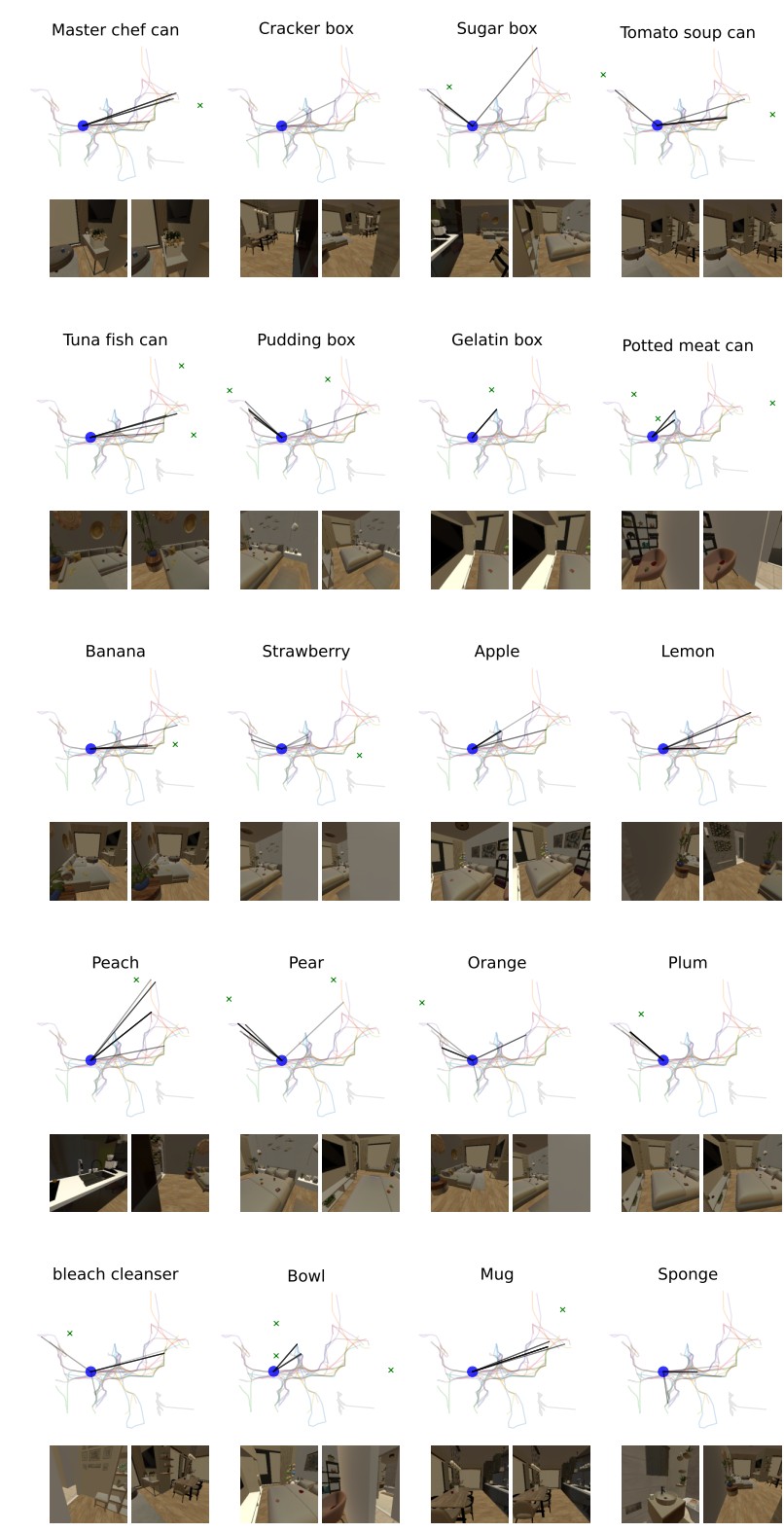

Figure 15: Attention scores of the object detection head described in Appendix C.2. The colored curves are the trajectories of previous episodes. The blue circle is the agent's position. The black lines represent the agent's attention when the target is the type in above the image. The lines connect the agent with the point in history that it attends to, the opacity of the line represents the attention score. The two images with highest attention score are shown in the 3rd row.