# OpenReview forum: "ReLIC: A Recipe for 64k Steps of In-Context Reinforcement Learning for Embodied AI"
_ICLR.cc/2025/Conference — Submitted to ICLR 2025_

### Official Review · Reviewer_pU2u · 2024-11-01

**Soundness:** 2
**Presentation:** 2
**Contribution:** 2
**Rating:** 5
**Confidence:** 3

**Summary:**

The paper introduces ReLIC, a framework designed to extend in-context learning capabilities to embodied AI agents. ReLIC focuses on enabling agents to utilize up to 64,000 steps of contextual experience, aiming to allow rapid adaptation in unseen environments, especially for multi-object navigation tasks. This is achieved through two components: (1) Partial Updates, a policy update scheme that incrementally adjusts the policy within rollouts, and (2) Sink-KV, a mechanism for efficient attention management in transformers, which optimizes long-sequence processing by adding learnable sink key-value vectors in attention layers.

**Strengths:**

1. The paper is generally well-written and easy-to-understand.
2. Experiments show that the proposed approach is effective.
3. Hyperparameters and algorithm details are provided for reproducity.

**Weaknesses:**

1. The two major novel components of the paper as claimed by the authors are the partial updates and Sink-KV. For the former one, if I understand correctly, it is a very common technique used in common RL training (I don't if it's the same case for in-context RL as I'm not familiar with its literature), namely, update the policy/value in the middle of an episode to increase training frequency. For Sink-KV, it seems to be interesting but I'm not familiar with the literature exploring the architecture of transformer so I don't how novel this is, but one thing that is not clear to me is that, why this design especially helps in-context RL training? What makes it special compared to using it in vision/language tasks?

2. For the experiments, the author mainly compared their method to meta-rl baselines and different variants of their method - while this is good, I would expect to see how the proposed method compared to other in-context RL baselines.

3. The emergent imitation learning results are not that surprising to me - a lot of recent work  do pretraining with imitation learning and then rl finetuning, the proposed scheme here is like first pretraining with rl then finetuning with imitation learning. This setting is also a little bit strange as in the real world case, if we have the expert demonstrations, pretraining on that usally greatly speed up RL.

4. std/error bar not included in the darkroom result plot and miniworld result plot

**Questions:**

See above

---

> ### Author Response · Authors · 2024-11-23
>
> We thank the reviewer for the comments and suggestions. We address the reviewer’s points below.
>
> **1. How does the Sink-KV especially help for in-context RL training compared to vision/language tasks?**
>
> Sink-KV is especially helpful for in-context RL since unlike in vision/language tasks, the agent takes actions in an environment to explore, with early exploration often providing little useful information. Without Sink-KV the agent will be forced to attend to these low information visuals. For example, in ExtObjNav, when the agent is first placed in an unseen house layout, the object can be anywhere in the house, meaning the agent needs to randomly explore at first. These random explorations through the house will have little new visual information. SInk-KV resolves the issue of requiring the attention scores over these low-information sequences to sum to 1 as in standard attention. Figure 3c empirically verifies the utility of Sink-KV and Figure 13 compares with alternative implementations. We moved the discussion from Appendix E.1 into the main paper to discuss Sink-KV in more detail.
>
> **2. The paper compares to Meta-RL baselines and variants of the method. How does it compare to other in-context RL baselines?**
>
> Our paper does compare to other in-context RL baselines since RL2 and Transformer-XL are both in-context baselines, which ReLIC greatly outperforms (Figure 2). In-context RL is a form of Meta-RL where the policy implicity infers the task context by taking an entire history of interactions as input (Section 2, L11-114). RL2 is an in-context RL method since it uses an LSTM to operate over the entire context of inputs. The “Transformer-XL” baseline does the same with a Transformer-XL sequence model, which is the same approach used in prior work for in-context RL [1].
>
> **3. The proposed scheme of first pretraining with RL and then finetuning with imitation learning is a bit strange. If we have expert demonstrations, training with them speeds up RL.**
>
> By training with RL, ReLIC avoids the need for vast quantities of expert demonstrations which for ExtObjNav do not already exist and are costly to obtain. For example, [2] spends significant resources obtaining 80k demonstrations in an ObjectNav task. Our results show in-context imitation learning with just several demonstrations, without having to finetune the model (Figure 4c). However, if demonstrations are available, ReLIC is still compatible with imitation learning. Imitation learning can first pretrain the model and then RL can finetune the model for in-context RL capabilities. Exploring how to combine imitation learning into ReLIC is an interesting direction for future work.
>
> **4. Emergent imitation learning results are not that surprising to me.**
>
> To the best of our knowledge, ReLIC provides the first empirical results that in-context imitation learning can emerge from only RL training. The in-context imitation learning we demonstrate provides expert demonstrations of navigating to objects in unseen houses via shortest path trajectories, and thus are new behaviors to teach the agent. The ReLIC policy generalizes to utilizing these demonstrations in-context to efficiently navigate to these objects, without any finetuning. Such shortest path demonstrations were not present in ReLIC RL training, which only utilized the environment reward.
>
> **5. Std/error bar not included in the Darkroom and Miniworld result plots.**
>
> Thank you for the suggestion, we updated Figure 5 to include the standard error as the error bars for ReLIC. ReLIC has small standard error in both environments.
>
> **References**
>
> [1] Team, Adaptive Agent, et al. "Human-timescale adaptation in an open-ended task space." 2023.
>
> [2] Ramrakhya, et al. "Habitat-web: Learning embodied object-search strategies from human demonstrations at scale." 2022.

---

> ### Author Response · Authors · 2024-12-03
>
> We thank the reviewer for their feedback. As the discussion period is near its conclusion, we would be grateful if you can comment on whether our response addressed your concerns or if issues remain.

---

### Official Review · Reviewer_tdez · 2024-11-03

**Soundness:** 3
**Presentation:** 2
**Contribution:** 2
**Rating:** 3
**Confidence:** 3

**Summary:**

* This paper proposes a new in-context RL approach with 64K historied steps. They introduce partial update schemes to solve sample inefficiency problems and Sink-KV mechanism to address long observation history.

**Strengths:**

*  The problems proposed by the paper are meaningful. Generalization is a long-term goal for embodied AI. In context learning with longer prompts may be a potential solution.
*  This paper is well-structured, conducts rich experiments and achieves great performance.

**Weaknesses:**

*  The description for "partial update" is confusing. Providing pseudocode would be good for understanding.
*  The contributions of the paper are confusing. This paper proposes two ways to solve long in-context RL: partial update and SINK-KV. "Partial update" is more like a training trick and "Sink-KV" is proposed by previous works. In the appendix, the paper explaints "Sink-KV" will help the policy for exploration. This is interesting and it should be place in the main paper.
*  Lack of some comparisons. Like transformer with multi episodes and DPT in ExtObjNav.
*  Environments used in experiments are too simple and they are all navigations tasks. It would be interesting to include manipulation benchmarks like MetaWorld.
*  Inefficient and large GPU memory for long context length. What is the inference speed for ReLIC and other baselines?
*  This paper emphasise 64K in the title and introduction. But in the experiment sections, all results are evaluated under small context length.

**Questions:**

*  See weakness.
*  In the Darkroom and miniwork experiment, all baselines have very low performance (0 success rate) with no context while ReLIC has higher performance (30% success rate). Is it normal? And DPT can get a very higher improvement with 20 in-context episodes.

---

> ### Author Response · Authors · 2024-11-23
>
> We thank the reviewer for the comments and suggestions. We address the reviewer’s points below.
>
> **1. Lacking some comparisons in ExtObjNav, like “DPT” and “transformer with multi-episodes”.**
>
> DPT is not possible to compare against in ExtObjNav because DPT requires an optimal policy to label the action sequences during learning and we do not have an optimal policy in ExtObjNav. Computing optimal actions in ExtObjNav is challenging because the task requires exploring an unseen house to find an object from egocentric RGB perception, without any exact object locations, object 3D geometry or a map of the environment. The DPT results in Darkroom and Miniworld use an optimal policy, but ReLIC does not. We updated Section 4.5 to clarify this additional DPT assumption.
>
> The “transformer with multi-episodes” approach is the same as ReLIC without partial updates or Sink-KV. Our results in Figure 3b,c also show that this is an infeasible approach, motivating the partial updates and Sink-KV in ReLIC. T However, Figure 3b shows that removing partial updates results in poor in-context learning. Likewise, Figure 3c shows that no Sink-KV results in far slower RL training. Thus both of these components are important for ReLIC.
>
> **2. Inefficient and large GPU memory for long context length.**
>
> During inference, the GPU memory of the KV cache for a batch size of 20 and sequence length 8200 is only 672MB, thus not prohibitively large. Despite being able to leverage long context lengths, the ReLIC policy itself is only 6.5M parameters, excluding the pretrained visual encoder which is not updated during training. Furthermore, since ReLIC is based on the standard Llama transformer architecture, it also benefits from inference speedups such as quantized KV caching [1] and training speedups like FlashAttention [2].
>
> **3. What is the inference speed for ReLIC and other baselines?**
>
> The inference speeds of ReLIC, Transformer-XL, and RL2 are listed in the table below. All numbers were obtained with batch size 20 on a single A40 GPU. The models are all about 5.5M parameters in size and operate on an 8k context length. They all have similar inference speeds with RL2 being faster due to its LSTM rather than transformer based architecture.
>
> |                | ReLIC   | Transformer-XL | RL2     |
> |----------------|---------|----------------|---------|
> | Actions per Second            | 732.14  | 777.25         | 893.73  |
>
> We updated Supplementary F to include this table.
>
> **4. This paper emphasizes 64K in the title and introduction. But in the experiment sections, all results are evaluated under small context length.**
>
> This is incorrect because our results in the main paper do show 64k context length. The paragraph on L412, titled “64k step trials” discusses results at 64k context length with the result in Figure 4b subtitled “64k Train Context Length”. These results demonstrate ReLIC can in-context learn with 64k steps of experience spanning over 175 episodes and continuing to improve the success rate with each new episode. We do not show baseline results for this longer context length setting because Figure 2 shows they fail to in-context learn for 8k context length.
>
> **5. In Darkroom and Miniworld, ReLIC starts at high performance, other baselines have low performance, and DPT quickly in-context learns. Is this normal?**
>
> Yes, this is expected because, as previously mentioned, DPT is trained with optimal action trajectories, yet is evaluated in Darkroom and Miniworld with agent generated actions. Thus, it initially achieves low performance. ReLIC starts with higher performance because it is trained with self-generated experience via RL, thus it performs well when evaluating the first episode, which corresponds to 0 in-context episodes in Fig. 5. The low starting performance results in DPT quickly in-context learning over the first 20 in-context episodes. However, ReLIC still reaches higher performance than all baselines with a lower number of ICL episodes.

---

> ### Author Response · Authors · 2024-11-23
>
> **6. Environments are too simple and all experiments are with navigation tasks.**
>
> We would like to highlight that ExtObjNav is a challenging problem since it requires finding small objects in large 3D homes from partially observable egocentric RGB observations and the agent must learn in-context in an entirely new house at test time. To illustrate the complexity of this task, observe that policies in ExtObjNav required almost a billion steps of training (see Fig 3a). This is over 2x more training with PPO required compared to the most complex MetaWorld setting, ML-45. Even the standard ObjectNav task that ExtObjNav extends is challenging, with prior work training policies with PPO for over 700M steps to learn searching for a single large furniture object [3], without the additional challenge of in-context learning over multiple episodes as in ExtObjNav. Furthermore, ExtObjNav benefits from over 175 episodes of in-context learning (Fig. 4b of the main paper), which amounts to 64,000 steps of experience, which at a 1Hz decision-making frequency is over 17 hours of in-context experience.
>
> **7. The description for "partial update" is confusing. Pseudocode would be good for understanding.**
>
> Thank you for the suggestion. We added pseudocode for the partial update to Section 3.3. Partial updates address the problem of long contexts of learning leading to sample inefficiency. In PPO, a sequence of experience is collected under the same policy, and then that policy is updated with this collected experience. However, with long contexts, this sequence would have to be prohibitively long before each policy update. Partial updates condition the policy on the entire context from a long sequence, but only apply the RL loss to the latest steps of the context since the last policy update, with the context persisting between updates. We show the importance of partial updates in ExtObjNav in Figure 3b.
>
> **7. In the appendix, the paper explains "Sink-KV" will help the policy for exploration. This is interesting and it should be placed in the main paper.**
>
> Thank you for the suggestion. We moved this explanation to Section 3.2 of the main paper to better explain and motivate the Sink-KV technique.
>
> **References**
>
> [1] Hooper, et al. "Kvquant: Towards 10 million context length llm inference with kv cache quantization." 2024.
>
> [2] Dao, et al. "FlashAttention: Fast and memory-efficient exact attention with io-awareness." 2022.
>
> [3] Khanna, et al. "Habitat synthetic scenes dataset (hssd-200): An analysis of 3d scene scale and realism tradeoffs for objectgoal navigation." 2024.

---

> ### Author Response · Authors · 2024-12-03
>
> We thank the reviewer for their feedback. As the discussion period is near its conclusion, we would be grateful if you can comment on whether our response addressed your concerns or if issues remain.

---

### Official Review · Reviewer_e3Wg · 2024-11-08

**Soundness:** 3
**Presentation:** 3
**Contribution:** 3
**Rating:** 6
**Confidence:** 3

**Summary:**

The paper introduces ReLIC (Reinforcement Learning In Context), a new approach that enables embodied AI agents to adapt to new environments using up to 64,000 steps of in-context experience. ReLIC addresses two key challenges in scaling in-context reinforcement learning: efficiently training with long context windows and effectively utilizing visual information from multiple episodes. To solve these challenges, the authors introduce "partial updates," a novel policy update scheme that allows more frequent learning during long rollouts, and "Sink-KV," a modification to transformer attention that helps the model selectively attend to relevant information in long sequences.

The authors evaluate ReLIC on an extended version of object navigation where agents must find multiple objects in unseen house layouts. ReLIC significantly outperforms baseline approaches, achieving a 43% success rate compared to 22% from the closest baseline after 15 episodes of in-context experience. The method also demonstrates an unexpected capability for few-shot imitation learning despite never being trained with expert demonstrations. Through extensive ablation studies, the authors show that both partial updates and Sink-KV are crucial for effective learning, and that sufficient training scale is necessary for in-context learning capabilities to emerge. The approach also performs well on existing benchmarks like Darkroom and Miniworld tasks, showing better and faster adaptation compared to prior work.

**Strengths:**

1. Strong Empirical Results: The method achieves substantial improvements over baselines, nearly doubling the success rate (43% vs 22%) in challenging navigation tasks. The results are thoroughly validated across multiple environments (EXTOBJNAV, Darkroom, Miniworld) with comprehensive ablation studies.
2. Scalability: The paper demonstrates impressive scaling capabilities, showing that their method can handle up to 64,000 steps of context - a significant improvement over previous approaches. More importantly, they show that models trained on shorter contexts (4k steps) can generalize to much longer contexts (32k steps) at inference time.

**Weaknesses:**

1. Limited Success Rate: Despite significant improvements over baselines, the absolute performance (43% success rate) is still relatively low for practical applications. The paper doesn't thoroughly discuss why this ceiling exists or potential paths to improve it.

2. Computational Cost: The method requires substantial computational resources - 12 days of training on 4 NVIDIA A40 GPUs. This high computational requirement could limit the practical applicability and accessibility of the approach, especially for researchers with limited resources.

3. Task Limitation: The evaluation focuses primarily on navigation tasks with discrete action spaces. There's no exploration of how the method might extend to continuous action spaces or more complex tasks like manipulation, limiting our understanding of the approach's broader applicability.

4. Missing Related work:
Retrieval-Augmented Decision Transformer: External Memory for In-context RL, Schmied et.al
	 - Work is quite similar to yours. It would be interesting if the authors could point out the differences in the related work section. This will improve the paper.

**Questions:**

1. Scalability and Practical Applications: Given the high computational requirements (12 days on 4 A40 GPUs), what approaches have you considered to make ReLIC more practically applicable? Could techniques like model distillation or more efficient architectures reduce these requirements while maintaining performance?
2. Beyond Navigation: The current work focuses on discrete action spaces in navigation tasks. Have you explored how ReLIC could be extended to continuous action spaces or more complex tasks like manipulation? What modifications would be needed to handle such scenarios?
3. Emergent Few-shot Learning: The emergence of few-shot imitation learning capabilities without explicit training is intriguing. What mechanisms in ReLIC enable this capability, and how could it be enhanced? Have you analyzed how the quality of demonstrations affects performance?

---

> ### Author Response · Authors · 2024-11-23
>
> We thank the reviewer for the comments and suggestions. We address the reviewer’s points below.
>
> **1. Despite significant improvements over baselines, the absolute performance is still relatively low for practical applications. The paper doesn't thoroughly discuss why this ceiling exists or potential paths to improve it.**
>
> The success rate of 43% is due to the challenging nature of the ExtObjNav task which requires searching for small objects in houses not seen during training. Our results are in line with previous works that do object navigation to large receptacles in the same HSSD scenes and achieve 48% success rate [1]. Overall, exploring an unseen house to find an object from egocentric RGB perception, without any exact object locations, object 3D geometry or a map of the environment is an overall challenging task.
>
> One path to improving this performance is to scale training with more RL training and in-context learning. Figure 4b shows the performance is still improving after 64k steps of in-context experience. Another path is to improve ability to generalize to new scenes by increasing the number of training scenes from the 37 in the HSSD dataset used in ExtObjNav. This could be accomplished by procedurally generating scenes like in [2].
>
> We added this discussion on future work in paths to improving performance to Section 5 of the paper.
>
> **2. The high computational requirement of ReLIC could limit the applicability and accessibility of the approach.**
>
> ReLIC’s high computational requirement is a consequence of the high computational demands of Embodied AI platforms used to simulate ExtObjNav, which is in line with prior work. Prior works show that RL in such 3D, house space environments typically use 8-64 GPUs per experiment [2,3,4], whereas ReLIC falls into this range with 4 A40 GPUs used for 12 days to reproduce our main results.
>
> We also measured the inference speeds of ReLIC, Transformer-XL and RL2 in the table below measured on batch size 20 on a single A40 GPU. Despite all methods operating with the same 8k context length and similar model parameter counts, they all have similar inference speeds with RL2 being faster due to its LSTM rather than transformer based architecture. This fast inference speed for ReLIC makes it more broadly applicable when decision-making latency is a concern.
>
> |                | ReLIC   | Transformer-XL | RL2     |
> |----------------|---------|----------------|---------|
> | Actions per Second            | 732.14  | 777.25         | 893.73  |
>
> We updated Supplementary F to include this table.
>
> **3. How to make ReLIC more computationally efficient? Could techniques like distillation or efficient architectures help?**
>
> ReLIC can be made more computationally efficient by utilizing the same transformer optimizations used in popular LLMs like Llama because ReLIC is based on the same architecture. Thus our method benefits from the same computational efficiency gains like flash attention [5] or quantized KV caching [6]. Distillation could also help with greater efficiency where a teacher policy could leverage oracle information about the scene such as ground truth object positions and scene layout and distill decision-making into a student ReLIC policy that operates from visual inputs.
>
> **4. How to extend ReLIC to continuous action spaces and complex tasks like manipulation?**
>
> In principle, ReLIC can be extended to continuous action spaces without modification. Nothing about it is specific to discrete control.
>
> We would like to highlight that ExtObjNav is a challenging problem since it requires finding small objects in large 3D homes from partially observable egocentric RGB observations and the agent must learn in-context in an entirely new house at test time.
>
> **5. Missing Related work: “Retrieval-Augmented Decision Transformer: External Memory for In-context RL, Schmied et.al”.**
>
> While this work appeared online after the ICLR deadline, meaning it was impossible to include in our submission, we briefly summarize the differences in this response. The RA-DT method from Schmied et al uses an external memory mechanism to operate over previous experiences from the context, while ReLIC simply operates over the prior experience as input in the context to a transformer. Despite ReLIC being an end-to-end approach without any explicit memory store, we show ReLIC scales to a longer context of 64k steps compared to the 6400 sequence length used in RA-DT.

---

> ### Author Response · Authors · 2024-11-23
>
> **6. What mechanisms in ReLIC enable few-shot learning and how can it be enhanced? How does the demonstration quality affect performance?**
>
> We hypothesize that few-shot imitation learning is enabled by the ReLIC training with diverse agent generated experiences. During RL training, the agent generates the experiences that are added to the context. At test time, this results in ReLIC generalizing to the expert demonstrations in the context during few-shot imitation learning. This capability could be enhanced by increasing the diversity of the experience during training to better generalize to new types of demonstration data in the context.
>
> Figure 4c shows that high-quality demonstrations from shortest-path trajectories improve learning speed and final performance over lower quality agent generated trajectories. However, future work can explore how this is affected by different types of demonstrations such as from human demonstrations.
>
> **References**
>
> [1] Khanna, et al. "Habitat synthetic scenes dataset (hssd-200): An analysis of 3d scene scale and realism tradeoffs for objectgoal navigation." 2024.
>
> [2] Deitke, et al. "ProcTHOR: Large-Scale Embodied AI Using Procedural Generation." 2022.
>
> [3] Yadav, Karmesh, et al. "Ovrl-v2: A simple state-of-art baseline for imagenav and objectnav." 2023.
>
> [4] Hu, et al. "Flare: Achieving masterful and adaptive robot policies with large-scale reinforcement learning fine-tuning." 2024.
>
> [5] Puig, et al. "Habitat 3.0: A co-habitat for humans, avatars and robots." 2023.
>
> [6] Hooper, et al. "Kvquant: Towards 10 million context length llm inference with kv cache quantization." 2024.
>
> [7] Dao, et al. "FlashAttention: Fast and memory-efficient exact attention with io-awareness." 2022.

---

> ### Author Response · Authors · 2024-12-03
>
> We thank the reviewer for their feedback. As the discussion period is near its conclusion, we would be grateful if you can comment on whether our response addressed your concerns or if issues remain.

---

### Author Response · Authors · 2024-12-03

We thank the reviewers for their insightful feedback and support. Reviewers highlighted that our work achieves strong empirical results (e3Wg,tdez,pU2u), is scalable (e3Wg), and is validated across different environments with comprehensive ablations and experiments (e3Wg,tdez,pU2u). We highlight that our rebuttal added the following:

**1. Details on computational efficiency (e3Wg,tdez):** We added inference speeds for ReLIC and baselines demonstrating that ReLIC achieves similar inference speeds as baselines in Supplementary F. We also highlighted that the compute required for training ReLIC with 4 A40 GPUs for 12 days is due to the simulation and is consistent with prior works in embodied AI, which typically use 8-64 GPUs per-experiment. Finally, we discussed how ReLIC is only 6.5M parameters and thus does not consume prohibitively large GPU memory.

**2. Explanation that the baselines in our experiments are comprehensive (tdez,pU2u)**. In ExtObjNav, comparing to DPT is infeasible because there is no optimal policy and ReLIC without Sink-KV and partial updates is unable to learn. We also clarified that ReLIC does outperform the other in-context RL baselines of RL2 and Transformer-XL.

**3. Clarified the Sink-KV and partial update methodology components of ReLIC (tdez,pU2u)** We added pseudocode for the partial update and updated the description in Section 3.3. We also added more details about how Sink-KV helps with embodied AI to Section 3.2.

---

### Meta-Review · Area_Chair_FYfr · 2024-12-18

**Metareview:**

The paper presents an approach for using large context transformers to attend to large amounts of experience. The reviewers have a split opinion on this paper, which was not fully resolved during the discussion phase. Hence I have taken a closer look at the paper. On the positive side, I tend to agree with reviewer e3Wg that using large context for transformers is quite interesting. However, the weaknesses of limited applicability is quite apparent. A key motivation of the paper is robotics. But it seems like there are no robot experiments even with relatively simple navigation tasks. It is hence unclear the larger impact the core ideas in the work can have.

**Additional Comments On Reviewer Discussion:**

Reviewers did not discuss much. I took a closer look at the paper and agree with the negative comments on broader applicability and evaluations being limited.

---

### Decision · Program_Chairs · 2025-01-22

Reject